# Comprehensive In Vitro and In Silico Aerodynamic Analysis of High-Dose Ibuprofen- and Mannitol-Containing Dry Powder Inhalers for the Treatment of Cystic Fibrosis

**DOI:** 10.3390/pharmaceutics16111465

**Published:** 2024-11-17

**Authors:** Petra Party, Zsófia Ilona Piszman, Árpád Farkas, Rita Ambrus

**Affiliations:** 1Institute of Pharmaceutical Technology and Regulatory Affairs, University of Szeged, Eötvös Street 6, 6720 Szeged, Hungary; party.petra@szte.hu (P.P.); piszman.zsofi@gmail.com (Z.I.P.); 2Centre for Energy Research, Hungarian Academy of Sciences, Konkoly-Thege Miklós Street 29-33, 1121 Budapest, Hungary; farkas.arpad@ek.hun-ren.hu

**Keywords:** dry powder inhaler, mannitol, ibuprofen, wet milling, spray drying, Andersen Cascade Impactor, stochastic lung model, cystic fibrosis

## Abstract

**Background:** Cystic fibrosis is a hereditary disease, which causes the accumulation of dense mucus in the lungs accompanied by frequent local inflammation. The non-steroidal anti-inflammatory drug ibuprofen (IBU) and the mucolytic mannitol (MAN) can treat these symptoms. Compared to per os administration, a lower dose of these drugs is sufficient to achieve the desired effect by delivering them in a pulmonary manner. However, it is still a challenge to administer high drug doses to the lungs. We aim to develop two inhaled powder formulations, a single-drug product of MAN and a combined formulation containing IBU and MAN. **Methods:** MAN was dissolved in an aqueous solution of Poloxamer-188 (POL). In the case of the combined formulation, a suspension was first prepared in a planetary mill via wet milling in POL medium. After the addition of leucine (LEU), the formulations were spray-dried. The prepared DPI samples were analyzed by using laser diffraction, scanning electron microscopy, powder X-ray diffraction, differential scanning calorimetry, density tests, in vitro aerodynamic studies (Andersen Cascade Impactor, Spraytec^®^ device), in vitro dissolution tests in artificial lung fluid, and in silico tests with stochastic lung model. **Results:** The DPIs showed suitability for inhalation with low-density spherical particles of appropriate size. The LEU-containing systems were characterized by high lung deposition and adequate aerodynamic diameter. The amorphization during the procedures resulted in rapid drug release. **Conclusions:** We have successfully produced a single-drug formulation and an innovative combination formulation, which could provide complex treatment for patients with cystic fibrosis to improve their quality of life.

## 1. Introduction

Cystic fibrosis (CF) is a hereditary monogenic disease that affects, amongst other organs, the lungs. The cause of the disease is a mutation in the cystic fibrosis transmembrane conductance regulator (CFTR) gene, which leads to the absence, loss, or impaired function of an ion channel in the lungs, named the CFTR protein. Consequently, viscous mucus leads to infections, chronic inflammation, and long-term lung destruction. In the past, CF was lethal for infants and young children, but the advancement of early diagnosis and treatment protocols has led to a median survival of 50 years of age today [1,2].

The primary aim of CF therapy is to maintain or even enhance the patients’ quality of life. Alternative forms of treatment include specific diets and airway clearance treatments, which do not involve the use of pharmaceuticals. Mucolytics, such as mannitol, dornase-alpha, and hypertonic sodium chloride solution, are also commonly utilized to dilute the thick mucus. Bacterial infections potentially leading to exacerbations can be treated with antibiotics. The most typical ways to provide these medications (such as azithromycin, colistin, tobramycin, and ciprofloxacin) are by intravenous, pulmonary, and oral routes. In addition to antibiotics, anti-inflammatory drugs, particularly corticosteroids, are helpful in such treatment; nevertheless, their side effects could be harmful in long-term therapies. CFTR modulators, such as Symdeko Trikafta, Orkambi, and Kalydeco, have significantly improved CF management and therapy. However, they are only suitable for specific mutations in the CFTR gene. These are the priciest medications, and before administering them to patients, a genetic examination is essential [3,4,5].

In treating CF, the most favorable noninvasive delivery option is administrating the drug directly into the lungs. With such a technique, a high local concentration of the drug can be achieved while lowering the systematic exposure; thus, a similar therapeutic effect with lower drug doses can be reached in comparison with oral administration [6]. Traditionally, there are three main options for delivery, such as metered-dose inhalers (MDIs), nebulizers, and dry powder inhalers (DPIs). They use fundamentally different techniques for drug delivery; therefore, the development of new formulations needs to be targeted toward one delivery method [7].

In recent years, DPI systems gained popularity due to being environmentally and user-friendly. They do not require propellants for their operations, consequently reducing their environmental impact [8,9]. The device is compact and portable, which is favorable for patient compliance [10]. The delivery of the powder is activated upon inhalation; thus, complex hand–lung coordination is not required. In comparison, the operation of MDIs requires such coordination; consequently, in many patients, the handling of DPIs requires less learning. However, it can vary by the exact DPI device used [11]. DPIs require a special inhalation technique and therefore can only be used by children aged five or six and up [12,13]. To achieve optimal lung deposition with DPIs, both a sufficient inhalation maneuver and the optimal type of DPI device are required. The inhalers with high resistance are favorable for patients with impaired lung function to efficiently aerosolize powders from the devices at low inspiratory flow [14]. However, inhaler resistance alone should not be used to determine dosing [15]. The fine-particle fraction of the inhaled powder, a key determining factor of deposition, is dependent on the inhaled flow rates. Besides age, the severity of the disease must also be considered. In CF patients, lung function reduces as the patient gets older. Therefore, the patients’ characteristics need to be assessed individually when considering DPIs as part of the therapy regime.

Ibuprofen (IBU) is a non-steroidal anti-inflammatory drug (NSAID) commonly used in pediatrics. High doses of IBU have been proven to be effective in slowing down the decline in lung function in patients with CF, especially in children. Despite its recognized benefits, the risk of potential side effects, such as gastrointestinal bleeding, limits the more common use of chronic IBU treatment [16]. For the treatment of CF, the estimated maximum daily dose of IBU is 300 mg for pulmonary drug delivery, which is in contrast to the maximum of 3 g per day of the oral dose [17,18]. By directly targeting the inflammation in the lung with pulmonary delivery, we could take advantage of the promising effects of IBU whilst maintaining a lower plasma concentration, which possibly leads to a lower risk of gastrointestinal side effects [17]. There is an emerging need for inhaled NSAID medications to reduce inflammation in the lungs, besides corticosteroids, that has not yet been fulfilled. However, the application of NSAIDs in pulmonary therapy is not common due to their possible bronchoconstrictive side effect. A few NSAID-containing DPIs are mentioned in the literature (e.g., IBU and ketoprofen) for CF as under development, but no commercially available formulations are yet released [19,20,21,22,23].

Mannitol (MAN) could also be beneficial in the therapy of CF. It is an osmotically active sugar alcohol, used as a mucolytic agent in CF, in the form of a DPI. In patients 6 years and older, mannitol was found to be well tolerated and effective in improving lung function; nevertheless, the studies only included patients who showed no signs of bronchial hyperactivity during the preliminary mannitol tolerance test (MTT) [24,25]. However, currently, the only authorized MAN-containing medication in the EU, Bronchitol, is indicated for only patients 18 years or older. The daily dose of medication is 400 mg, divided into ten capsules, which could result in disadvantageous patient compliance [26]. Furthermore, MAN is an attractive excipient due to being less hygroscopic than some of the other sugars and not absorbing moisture until the RH is over 90%, which is beneficial in terms of better physical and chemical stability of the DPI formulation [27,28,29,30].

It is challenging to administer IBU and MAN locally since the medications’ recommended doses are high. For large drug dose delivery, carrier-free DPIs are preferred [23]; additionally, excipients with low Peclet numbers should be employed to execute optimal drug delivery and maintain a suitable capsule emptying process [31]. These developments would benefit patients since they may reduce the frequency of taking the medication. As a result, improved adherence to long-term treatment may be achieved [32]. As an additive, we used the Poloxamer-188 (POL) polymer, which is safe at a low dose and can help to create porous particles [33,34]. Moreover, POL coatings present on the particle surface could diminish the extent of biophysical inhibition of lung surfactants [35]. A further excipient, L-Leucine (LEU), was applied to enhance the dispersity of the particles, thereby improving the aerosolization and decreasing the hygroscopicity of the powder [36,37]. Moreover, LEU also has a stability-enhancing effect, mitigating moisture absorption with the hydrophobic shell [38,39].

Our goal was to develop a DPI containing only MAN as its active ingredient and a combined DPI containing both IBU and MAN to fill the gap in the development of carrier-free NSAID-containing DPIs. We expected the final DPI formulations to have high drug loading, a spherical shape, micro-sized particles, rapid active ingredient release, and suitable aerodynamic properties. In addition to MAN and IBU, the polymer Poloxamer 188 was used as an excipient to stabilize the microparticles, and the amino acid leucine was employed to improve the aerodynamic properties. The physiochemical properties of the finished products were formally evaluated using standard methods found in the European Pharmacopoeia, self-developed methods, and in silico aerodynamic investigations to comprehensively determine the pulmonary efficiency of the products.

## 2. Materials and Methods

### 2.1. Materials

The active pharmaceutical ingredient (API) was ibuprofen (IBU) (Sanofi, Veresegyház, Hungary). Poloxamer 188 (POL), (Sigma-Aldrich Co., Ltd. Budapest, Hungary), D-Mannitol (MAN) (Molar Chemicals Kft, Halásztelek, Hungary), and L-leucine (LEU), (AppliChem GmbH, Darmstadt, Germany) were chosen as excipients.

### 2.2. Methods

#### 2.2.1. Preparation of the MAN Solution and the IBU Presuspension

The MAN single-drug formulation was prepared by dissolving MAN, POL, and LEU in different concentrations. These components share a proper water solubility and mix well in the applied concentrations (Table 1).

In the case of the IBU-containing formulation, firstly, POL was dissolved in purified water, which resulted in a solution with a 1.0% (*w*/*w*%) concentration. POL is a polymer which prevents the aggregation of the drug particles during the particle size reduction method. This was followed by the preparation of a presuspension, which contained 2.00 g of pure IBU and 18.0 g of 1.0% POL solution, as a dispersant. The milling medium was 20.00 g of ZrO_2_ beads in a high-performance planetary ball mill (Fritsch Planetary Micro Mill Pulverisette 7, Fritsch GmbH, Idar-Oberstein, Germany). The optimized milling parameters were as follows: a rotation speed of 800 rpm, a milling time of 15 min, and a 10 min pause (4 cycles).

#### 2.2.2. Spray Drying of the Solution of MAN and the Presuspension of IBU

As a second step, DPI systems were produced with the help of a mini spray dryer, using the same parameters for both preparations. The ratio of the ingredients in the preparations was chosen based on the therapeutic doses and literature data (Table 1). Three different formulations were produced from the MAN solution and three different compositions were formulated from the IBU suspension by adding MAN and various amounts of LEU. The dry material contents of the final formulations are shown in Table 1. LEU is an amino acid, which enhances the dispersity of the spray-dried powders during aerosolization. A magnetic stirrer was used for its homogenization in the suspension (AREC.X heating magnetic stirrer, Velp Scientifica Srl, Usmate Velate, Italy). The inhalable powders were produced with a Büchi Mini Spray Dryer (Büchi Mini Spray Dryer B-191, Büchi, Flawil, Switzerland). Based on our preliminary experiments, the spray-drying settings were as follows: an inlet temperature of 70 °C, an aspirator capacity of 85%, an airflow rate of 500 L/h, and a pump rate of 3 mL/min.

#### 2.2.3. Preparation of the Physical Mixtures

Three physical mixtures (PMs) were prepared from the initial materials. Their compositions were equivalent to the spray-dried samples (Table 1). During the investigations, the properties of the physical mixtures were compared to the spray-dried products.

#### 2.2.4. Determination of the API Content

The API contents (Table 2) of the IBU-containing DPIs were determined by dissolving 1.0 mg of powder in 25 mL of methanol and a pH 7.4 phosphate buffer (90 + 10 *v/v*%) and were analyzed by UV/Vis spectrophotometry at a wavelength of 222 nm (Table 1). Measurements were carried out in triplicate.

#### 2.2.5. Laser Diffraction-Based Particle Size Measurement

Laser diffraction was used to determine the particle size distribution, and the specific surface area of our samples (Malvern Mastersizer Scirocco 2000, Malvern Instruments Ltd., Worcestershire, UK). In both cases, the refractive index of MX was adjusted to 1.55. The wet dispersion unit was used to investigate the particle size of the suspension. The suspension was measured in purified water while stirring at 2000 rpm. The dry dispersion unit was used to observe the spray-dried powders. The dispersion air pressure was set to 3.0 bar and a 75% vibration feed was applied. Each sample was measured in triplicate. The particle size distribution (PSD) was characterized by the values of D[0.1] (10% of the volume distribution is below this value), D[0.5] (50% of the volume distribution is below this value), and D[0.9] (90% of the volume distribution is below this value). Span values were revealed in the particle size distribution; the higher the Span value, the broader the distribution [40]. The specific surface area (SSA) was derived from the PSD data. The calculations were made under the assumption of spherical particles.

#### 2.2.6. Scanning Electron Microscopy Investigation

Scanning electron microscopy (SEM) (Hitachi S4700, Hitachi Scientific Ltd., Tokyo, Japan) was used to define the morphology of the spray-dried formulations. The investigation conditions were as follows: 10 kV high voltage, 10 mA amperage, and 1.3–13.1 mPa air pressure. A high vacuum evaporator and argon atmosphere were applied to make the sputter-coated samples conductive with gold palladium (Bio-Rad SC 502, VG Microtech, Uckfield, UK). For the implementation of the particle diameter investigation, ImageJ, a public domain image analyzer software, was used (ImageJ 1.53e, https://imagej.net/ij/ accessed on 30 September 2024).

#### 2.2.7. Density and Powder Rheology Measurement

The bulk and tapped densities of the formulations were measured using a tap density tester (ETD-1020x, Electrolab, Mumbai, India) [41]. A cylinder was filled with 1.5–2.0 cm^3^ of powders to calculate the bulk density (ρ_b_). It was tapped 1000 times. The tapped density (ρ_t_) was calculated compared to the volume of the powder before and after the taps. The measurements were performed three times. The Hausner ratio (HR) and Carr index (CI) values of the samples were evaluated from the bulk density and the tapped density (Equations (1) and (2)) [42]:(1)$$\mathrm{HR}=\frac{\rho_{t}}{\rho_{b}}$$
(2)$$\mathrm{CI}=\frac{\left(\rho_{t}-\rho_{b}\right)}{\rho_{t}}\times 100$$

#### 2.2.8. X-Ray Powder Diffraction Analysis

For the characterization of the crystalline materials, X-ray powder diffraction (XRPD) spectra were recorded with the help of the BRUKER D8 Advance X-ray diffractometer (Bruker AXS GmbH, Karlsruhe, Germany). The radiation source was Cu Kλ1 radiation (λ = 1.5406 Å). The parameters of the analysis were as follows: a Cu target, a Ni filter, a 40 kV voltage, a 40 mA current, a time constant of 0.1 °/min, and an angular step of 0.010° over the interval of 3–40°. DIFFRACT plus EVA 28 software (Bruker AXS GmbH, Karlsruhe, Germany) was used for the evaluation.

#### 2.2.9. Differential Scanning Calorimetry Investigation

Thermoanalytical properties were determined by differential scanning calorimetry (DSC). The measurements were executed with a Mettler Toledo DSC 821e thermal analysis system using the STARe thermal analysis program V9.1 (Mettler Inc., Schwerzenbach, Switzerland). Approximately 2–5 mg of the samples was observed in the temperature range between 25 °C and 300 °C. The heating rate was 10 °C/min. The carrier gas was argon at a flow rate of 10 L/h during the investigations.

#### 2.2.10. In Vitro Aerodynamic Measurements

The aerosolization properties of the spray-dried formulations were assessed in vitro using an Andersen Cascade Impactor (ACI) (Apparatus D, Copley Scientific Ltd., Nottingham, UK) [43]. The inhalation flow rate was set to 60 L/min (High-capacity Pump Model HCP5, Critical Flow Controller Model TPK, Copley Scientific Ltd., Nottingham, UK). The actual flow rate through the impactor was measured by a mass flow meter (Flow Meter Model DFM 2000, Copley Scientific Ltd., Nottingham, UK). The inhalation time was 4 s. The setting models the normal breathing pattern with a 4 l inhalation volume. Breezhaler^®^’s single-dose devices (Novartis International AG, Basel, Switzerland) were applied, with transparent, size-3 hydroxypropyl methylcellulose capsules (Ezeeflo™, ACG-Associated Capsules Pvt. Ltd., Mumbai, India). The capsule was filled with 10 mg of MAN and IBU-containing powders. To simulate the pulmonary adhesive circumstances, the collection plates on the stages were coated with Span 85 and cyclohexane (1 + 99 *w*/*w*%) mixture. In the case of the API-free samples, the weight of the device, the capsules, the induction port, the plates, and the filter were measured before and after inhalation. In the case of IBU-containing DPIs, the device, the capsules, the induction port, the plates, and the filter were washed with methanol and a pH 7.4 phosphate buffer (10 + 90 *v/v*%) to collect and dissolve the deposited amount of IBU. The API was quantified by UV/Vis spectrophotometry (ATI-UNICAM UV/VIS Spectrophotometer, Cambridge, UK) at a wavelength of 222 nm. The in vitro aerodynamic properties were evaluated with the help of Inhalytix™ 2.0.6. (Copley Scientific Ltd., Nottingham, UK) data analysis software, which is a validated aerodynamic particle size distribution data analysis program. The fine-particle fraction (FPF) and median mass aerodynamic diameter (MMAD) are the most widely used values. The FPF is defined as the percentage of the mass of the active ingredient consisting of particles with an aerodynamic diameter of fewer than 5 μm divided by the emitted dose of the formulations. The MMAD is influenced by the inhalation flow rate, density, size, and shape of the particle. The emitted fraction (EF) was also calculated, which is the released fraction from the DPI device.

#### 2.2.11. Aerodynamic Particle Size Analysis Using the Spraytec^®^ Device

The aerodynamic diameter was determined using a Spraytec^®^ laser diffractometer equipped with an inhalation cell (Malvern Instruments Ltd., Worcestershire, UK) and ACI. The investigation accounts for the EF of the DPI formulation and measures PSD directly from the inhalation device. SPD formulations were aerosolized from a cellulose capsule inserted into a Breezhaler^®^ device connected to an induction port of the inhalation cell. The assembly was attached to an ACI, which created a closed system that allowed for the measurement of the aerodynamic particle size under controlled circumstances [44]. The inhalation flow rate was set at 60 L/min. The inhalation time was 4 s. Measurements were made in triplicate.

#### 2.2.12. In Vitro Dissolution Test in Simulated Lung Media

Currently, there are no regulatory requirements for in vitro dissolution testing of inhaled products [45,46,47]. A modified paddle method (Hanson SR8 Plus, Teledyne Hanson Research, Chatsworth, CA, USA) by the European Pharmacopeia [48] was used to define the release of IBU from the solid dosage form. There is no optimal method to determine the exact volume of lung lining fluid. The estimated value is between 10 and 70 mL [49]. Considering the limitation of the dissolution setup, 50 mL of the previously mentioned (Section 2.2.10.) simulated lung medium was applied during the measurement [50,51]. The paddle was rotated at 100 rpm to continuously homogenize the media. The measurement was performed for up to 60 min at 37 °C [52]. A total of 5 mL of the samples was taken out after 5, 10, 15, 30, and 60 min. The medium was replenished in every case. After filtration (pore size: 0.45 µm, Millex-HV syringe-driven filter unit, Millipore Corporation, Bedford, MA, USA), the dissolved quantity of IBU was determined spectrophotometrically at a wavelength of 222 nm (ATI-UNICAM UV/VIS Spectrophotometer, Cambridge, UK). The measurement was executed three times.

#### 2.2.13. In Vitro Dissolution Test Using Paddle Method Combined with ACI

The release profile of IBU was determined from the respirable fraction of the formulations, which is recommended for better in vivo correlation [44]. The inhalation flow rate was set at 60 L/min. The inhalation time was 4 s. The stages of the ACI were modified, as a Fast-Screening Impactor to collect particles between 1 and 5 μm [50]. Between stages number one and four, a plate was applied covered with a polycarbonate (PCTE) membrane filter (Sterlitech, Auburn, WA, USA). On the last stage, a filter (A/E glass fiber filter, Pall Corporation, NY, USA) was applied to catch the smallest particles. A mass of each DPI formulation was filled into a cellulose capsule of size 3. A Breezhaler^®^ device was used for actuation. After inhalation, the filters were individually fixed on a watch glass–PTFE disk assembly (Copley Scientific Ltd., Nottingham, UK) with clips and a PTFE mesh screen [45]. The disk assembly was then immersed in a dissolution vessel of the Hanson SR8 Plus dissolution apparatus with 400 mL of artificial lung fluid [26,28]. The measurement was carried out for up to 60 min at 37 °C and the paddle was rotated at 100 rpm. Samples of 2 mL were taken after 5, 10, 15, 30, and 60 min. The medium was replenished in all cases. After filtration (pore size: 0.45 µm, Millex-HV syringe-driven filter unit, Millipore Corporation, Bedford, MA, USA), the dissolved quantity of IBU was determined spectrophotometrically at a wavelength of 222 nm (ATI-UNICAM UV/VIS Spectrophotometer, Cambridge, UK). The dissolution tests were performed in triplicate for each DPI formulation.

#### 2.2.14. In Silico Aerodynamic Characterization

The in silico simulations were performed using the stochastic lung model, which tracks the inhaled particles until their deposition or exhalation and computes the fraction of the particles deposited in each anatomical part of the airways [53]. The particle trajectories were simulated in an asymmetrical branching airway structure, mimicking realistic airways by selecting adequate morphometric parameters [54]. The input of the computational model can be different parameters characterizing aerosol particles like density, shape, or size, as well as the breathing parameters of the patient. In our work, the aerodynamic PSD of the samples measured by the ACI served as input for the numerical model of airway deposition. The inhalation parameters corresponded to the inhalation of a COPD patient through Breezhaler^®^, whose inhaled volume (IV = 1.7 L) and inhalation time (t_in_ = 1.6 s) corresponded to the best flow rate of the impactor measurements. Two different (5 s and 10 s) breath holding times were used. The computational deposition model was validated in earlier works [55,56].

## 3. Results

### 3.1. Result of Particle Size Analysis by Laser Diffraction

As a result of wet milling, the initial diameter of the API (D[0.5] = 31.78 ± 9.01 μm) was successfully decreased to 3.39 ± 1.51 (Table 2) in the presuspension of IBU. After spray drying, the D[0.5] values of the samples were between 3 and 8 μm. The result met the requirements of the inhaled powders in the case of LEU-containing products. Particle size is one of the most important factors in achieving sufficient lung deposition. Incorporating LEU, the geometric size of the spray-dried particles decreased, which led to an increasing SSA. According to the Span values, their particle size distribution was monodispersed (Span < 2.0), as a higher Span value results in a broader distribution [40]. When IBU was present in the systems, the distribution was only monodispersed in the case of the highest LEU-containing sample, which is crucial for accurate dosing [57].

### 3.2. The Outcomes of the Density and Powder Rheology Test

Low-density DPIs may provide a better profile of deposition in the airways. The ρ_t_ of the products was lower than or around 0.3 g/cm^3^. The density of the commercially available DPIs is approximately 1 g/cm^3^; therefore, the samples can be considered low-density formulations. Moreover, if the ρ_t_ is lower, the FPF will consequently be higher [58]. Aerosolization is additionally influenced by powder rheology properties, which can be determined by the HR and CI values (Table 3). The result indicate poor flowability, but these are similar to other carrier-free formulations in the literature [59,60].

### 3.3. Findings of Scanning Electron Microscopy Investigation

According to the morphological investigation of the particles, a nearly spherical shape was observed (Figure 1), which was the result of optimized spray drying and the presence of MAN [20,61]. Particle diameter was measured based on the SEM pictures with the help of the Image J program. The diameters were 10.51 ± 7.04 µm for POL_MAN2, 2.27 ± 0.99 µm for POL_MAN2_LEU0.5, and 1.95 ± 1.15 µm for POL_MAN2_LEU1. The data are the means ± SD (*n* = 100 independent measurements). Smooth surfaces are not preferred for pulmonary delivery since they tend to increase the interaction between particles, while rough or wrinkled surfaces tend to increase the aerosolization efficiency. When LEU was present in the systems, preferable wrinkled particles were established. These particles were forecasting a proper powder dispersion during inhalation, therefore resulting in higher drug delivery into the deeper regions of the lung [62,63,64]. Interestingly, a porous structure was observed in the samples containing both IBU and LEU, whose structure could also be beneficial during inhalation [65]. The diameters were 3.86 ± 1.43 µm for IBU1_POL_MAN2, 1.94 ± 1.09 µm for IBU1_POL_MAN2_LEU0.5, and 2.93 ± 1.46 µm for IBU1_ POL_MAN2_LEU1. The data are the means ± SD (*n* = 100 independent measurements).

### 3.4. Crystallinity Results of Particles Determined by X-Ray Powder Diffraction

The XRPD pattern of the physical mixtures demonstrated that MAN and LEU originally had a crystalline structure. The presence of POL did not affect the diffractograms, because it had no crystalline properties. In the case of the spray-dried products, the intensity of the characteristic peaks decreased (Figure 2). Overall, the spray-drying procedures decreased the number of crystallin particles of MAN and IBU. Initially, IBU also had a crystalline structure, and when compared to the physical mixture, the formulations’ characteristic peaks were also reduced. (Figure 3). After wet milling and spray drying, the API became partially amorphous, which can promote a better dissolution profile.

### 3.5. Findings of Thermoanalytical Investigation by Differential Scanning Calorimetry

DSC was applied to determine the melting of MAN and LEU in the physical mixtures and in the products (Figure 4). POL had no endothermic peak. MAN and LEU had an endothermic peak around 167 °C and 240 °C, respectively. The curves after solidification showed a lower melting point, which strengthened the findings of the XRPD investigation, the partial amorphization. The melting point of the IBU is observable in Figure 5. The API showed a sharp peak around 78 °C, reflecting its melting point and crystalline structure. After the preparation method, the DSC curves showed broader endothermic peaks of IBU, indicating a decrease in its crystallinity, just as the XRPD measurements have previously shown. The residual crystals melted at a lower temperature due to the smaller particle size and the amorphization.

### 3.6. Determined In Vitro Aerodynamic Properties of DPI Formulation

According to the outcomes of the laser diffraction analysis, the LEU-containing formulations were determined during the aerodynamic assessment. The deposition of the samples on different parts of the ACI is shown in Figure 6. The products partially remained in the capsule and a high amount was deposited in the induction port, which was expected due to the poor HR and CI values. Additionally, the aerodynamic performance might have decreased due to the increasing quantity of powder inside the capsule [66]. A small amount of the product reached the filter; therefore, it is less likely to be exhaled. The calculated in vitro aerodynamic results by using Inhalytix™ software are presented in Table 4. The MMAD values were between 3.8 and 4.8 µm, which is within the required size range for pulmonary delivery. The FPF results of the samples were between 61 and 71%, which is higher than the FPF values of the commercially available Breezhaler^®^ formulations [67] and other IBU-containing DPI formulations under development [68]. To reduce the amount of powder remaining in the capsules, the EF could be enhanced. However, the limitations of the ACI may be the cause of the resulting EF values. In addition to the stages, the other areas of the ACI also received powder deposition from the highly filled capsules. In CF, the right upper lobe is the most likely to develop inflammation and bronchiectasis; in comparison to this, in the lower lobes, mucus plugging and air trapping are more frequent [69,70]. The aim is to cover the entire lung, enabling access to both peripheral and central areas with the DPI. Based on these results, all samples containing LEU proved to be suitable for pulmonary delivery in CF.

### 3.7. Results of Aerodynamic Characterization by Spraytec^®^ Device

LEU-containing formulations were further analyzed by using a Spraytec^®^ device. The aerodynamic diameter values were similar to the ACI results (Table 5). However, only the lower LEU-containing samples were under 5 µm; therefore, this composition would be preferable for inhalation delivery. When compared to the geometric diameter results of the previously mentioned laser diffraction experiments, the aerodynamic diameter results are larger. This observation may be explained by the fact that in the case of a more porous structure, the particles are more likely to aggregate during aerosolization.

### 3.8. Outcomes of In Vitro Dissolution Investigation

The quantity of IBU released was minimal from the samples containing raw materials due to the inadequate water solubility of the drug (Figure 7). In the first 5 min, 63.69 ± 3.09% of the IBU was released from IBU1_POL_MAN2_LEU0.5 and 78.29 ± 0.69% from IBU1_POL_MAN2_LEU1. Moreover, 100% of the drug was released within an hour. The spray-dried samples showed significantly enhanced drug release compared to the physical mixtures. These improvements could be related to the higher specific surface area of the IBU compared to the raw drug and the partial amorphization of the IBU. The application of LEU caused a more porous structure; therefore, a larger amount of IBU was liberated from the powder.

### 3.9. Statements of In Vitro Dissolution Test Combined with ACI

The released amount of IBU was determined from the respirable fraction collected by the membrane applied in the ACI. The results were correlated with the ACI test and the conventional dissolution test (Figure 8). Due to its improved aerodynamic performance, the IBU1_POL_MAN2_LEU0.5 formulation provided better results in comparison to the IBU1_POL_MAN2_LEU1 powder. In conclusion, before in vivo measurements, it is critical to observe drug release from the fraction of the DPI, enhancing the accuracy of in vitro measurements, to determine the drug dose that is potentially delivered to the lung.

### 3.10. Results of In Silico Aerodynamic Characterization

During the in silico characterization of the deposited and exhaled fractions of the samples, different breath holding times were determined (Table 6). The extrathoracic deposition is lower for MAN-only-containing products. A total of 42 and 51% of the spray-dried formulations were deposited in the lungs. Using a breath holding time of 10 s, the deposition in the upper airways and the exhaled fraction decreased; therefore, the bronchial and acinar deposition improved in all cases. It was shown that the length of the breath holding time had a significant impact on the deposited fraction [71], which is not considered during in vitro aerodynamic evaluation. Therefore, the combination of in vitro and in silico techniques is advantageous to provide a more informative analysis, which is practical before clinical trials. Teaching the patients proper inhalation and breath holding techniques could improve the efficiency of the DPI.

## 4. Discussion

This study showed that a single-drug DPI preparation containing MAN and an innovative DPI combining MAN and IBU were developed using appropriate excipients for the treatment of CF. POL promoted the creation of more porous particles, and it could be beneficial for the reduction in the negative effects of the surfactant. The application of MAN, besides its mucus-diluting effect, led to a preferable spherical form. Using LEU, the particle surface became rough and more hydrophobic, which helped the aerodynamic properties and led to improved stability. In the case of IBU, the application of high doses (2–3 mg/kg/dose twice daily) is recommended for local CF treatment. According to the outcome of wet milling and spray drying, the formulation seemed consistent with the initial aim that high drug loading can be achieved, which can result in high-dose drug delivery to the lung.

A comprehensive physicochemical and dosage investigation was executed to determine the pulmonary applicability of the prepared samples. The particle size of the products was in the range that we predicted; the diameter of the formulations containing LEU was under 5 μm therefore they could target the lungs (POL_MAN2_LEU0.5_spd: 2.96 ± 0.05 μm; POL_MAN2_LEU1_spd: 3.28 ± 0.08 μm; IBU_POL_MAN2_LEU0.5_spd: 3.38 ± 0.13 μm; POL_MAN2_LEU1_spd: 3.28 ± 0.10 μm) according to the laser diffraction analysis. The MAN-only-containing formulation showed a spherical form, and the IBU-containing formulations appeared porous on the SEM pictures. Our findings suggest that this unique, porous shape was caused by the combined use of IBU and POL. Furthermore, the aerodynamic characteristics were found to be sufficient. The MMAD results of the LUE containing formulations were also in the required range for pulmonary delivery (POL_MAN2_LEU0.5_spd: 4.42 ± 0.15 μm; POL_MAN2_LEU1_spd: 4.71 ± 0.05 μm; IBU1_POL_MAN2_LEU0.5_spd: 3.88 ± 0.19 μm; IBU1_POL_MAN2_LEU1_spd: 4.76 ± 0.23 μm). FPF values were larger in comparison to the commercially available products in a Breezehaler^®^ device (POL_MAN2_LEU0.5_spd: 70.89 ± 8.59%; POL_MAN2_LEU1_spd: 64.34 ± 10.65%; IBU1_POL_MAN2_LEU0.5_spd: 63.62 ± 2.99%; IBU1_POL_MAN2_LEU1_spd: 61.91 ± 1.55%). The measurements with the laser diffraction-based Spraytec^®^ device equipped with an inhalation cell clarified that the lower LEU-containing product showed a preferable aerodynamic particle size for targeting the lung (POL_MAN2_LEU0.5: 4.24 ± 0.36 μm; POL_MAN2_LEU1: 6.02 ± 0.66 μm; IBU1_POL_MAN2_LEU0.5: 4.16 ± 0.13 μm; IBU1_POL_MAN2_LEU1: 5.12 ± 0.40 μm). In agreement with our original concept, from the micronized IBU-containing samples, the drug dissolved more quickly (69 ± 3.09% from IBU1_POL_MAN2_LEU0.5 and 78.29 ± 0.69% from IBU1_POL_MAN2_LEU1 within the first 5 min) than from the raw material-containing physical mixtures due to the decreased particle size of the drug reached by high-performance wet milling. In accordance with the ACI and Spraytec^®^ results, the combined dissolution test also proved that IBU1_POL_MAN2_LEU0.5 is the more beneficial formulation in comparison to IBU1_POL_MAN2_LEU1. Using the in silico method, we predicted the deposition of the formulations in human airways. According to the stochastic lung model, a large percentage of the formulations reached the bronchial and acinar parts of the lung (between 42 and 51%) and the prolonged breath holding time decreased the amount of the exhaled fraction. To examine the impact of varying relative humidities and temperatures on the crystalline structure of the APIs and excipients, it will be crucial to carry out a stability test of the presented DPIs in the future. In addition to having a therapeutic effect, the composition of our single-drug preparation can serve as a base for DPI formulations developed with other APIs in the future.

In conclusion, the combination of various excipients and APIs was observed, particularly from the perspective of particle size and shape. A comprehensive aerodynamic analysis was presented, which included various in vitro studies to measure the aerodynamic diameter and evaluate drug dissolution, as well as an in silico method to validate our predictions for the human body. The results can contribute to expanding our knowledge on high-dose DPI formulations, thereby providing the basis for future research of DPI formulations for the potential treatment of CF patients.

## Figures and Tables

**Figure 1 pharmaceutics-16-01465-f001:**
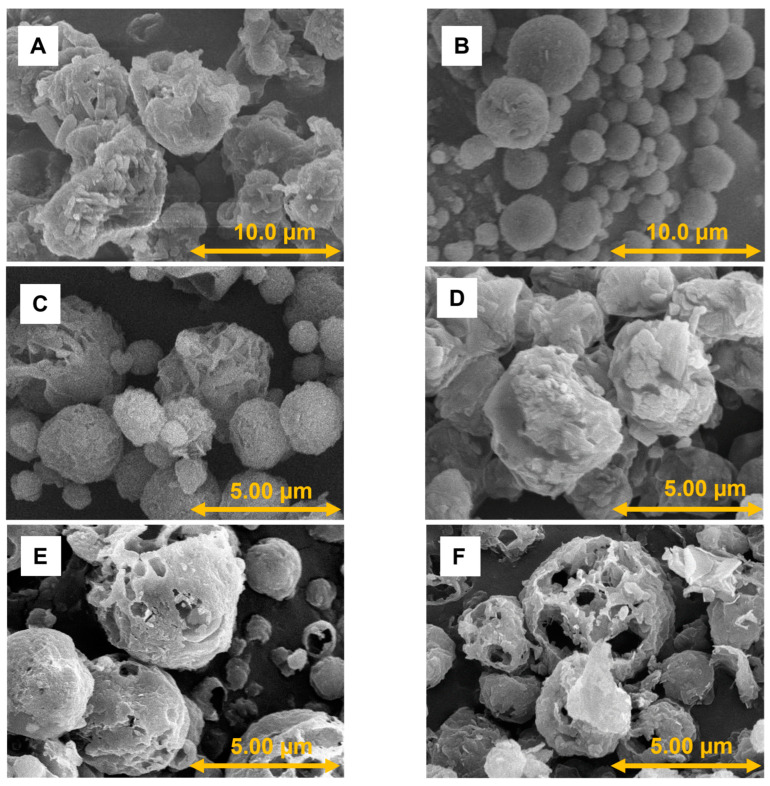
SEM pictures of the spray-dried samples: (**A**) POL_MAN2_spd, (**B**) POL_MAN2_LEU0.5_spd, (**C**) POL_MAN2_LEU1_spd, (**D**) IBU1_POL_MAN2_spd, (**E**) IBU1_POL_MAN2_LEU0.5_spd, and (**F**) IBU1_POL_MAN2_LEU1_spd.

**Figure 2 pharmaceutics-16-01465-f002:**
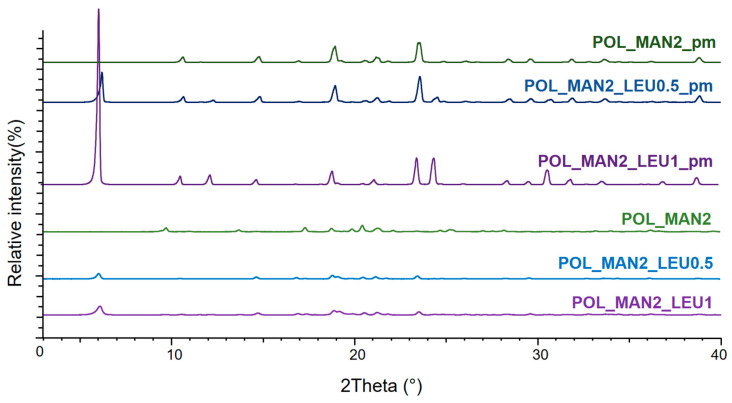
The XRPD results of the physical mixtures (POL_MAN2_pm, POL_MAN2_LEU0.5_pm, POL_MAN2_LEU1_pm) and the spray-dried MAN-containing samples (POL_MAN2_spd, POL_MAN2_LEU0.5_spd, POL_MAN2_LEU1_spd).

**Figure 3 pharmaceutics-16-01465-f003:**
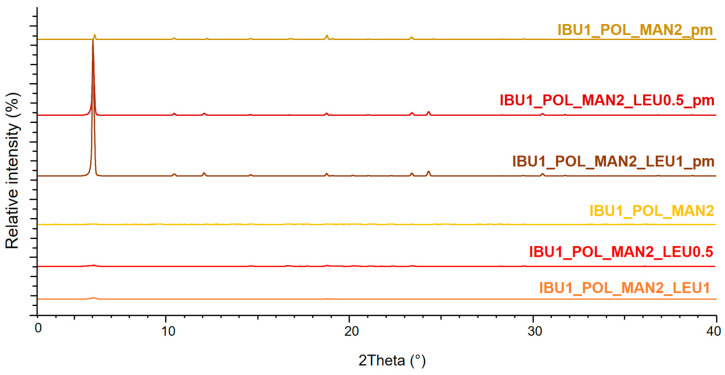
The XRPD results of the physical mixtures (IBU1_POL_MAN2_pm, IBU1_POL_MAN2_LEU0.5_pm, IBU1_POL_MAN2_LEU1_pm) and the spray-dried IBU- and MAN-containing samples (IBU1_POL_MAN2_spd, IBU1_POL_MAN2_LEU0.5_spd, IBU1_POL_MAN2_LEU1_spd).

**Figure 4 pharmaceutics-16-01465-f004:**
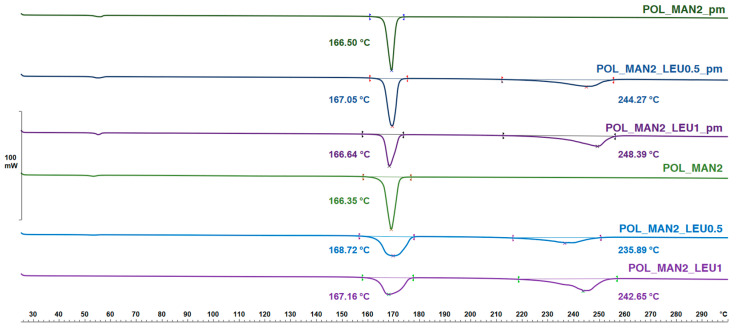
The DSC results of the physical mixtures (POL_MAN2_pm, POL_MAN2_LEU0.5_pm, POL_MAN2_LEU1_pm) and the spray-dried MAN-containing samples (POL_MAN2_spd, POL_MAN2_LEU0.5_spd, POL_MAN2_LEU1_spd).

**Figure 5 pharmaceutics-16-01465-f005:**
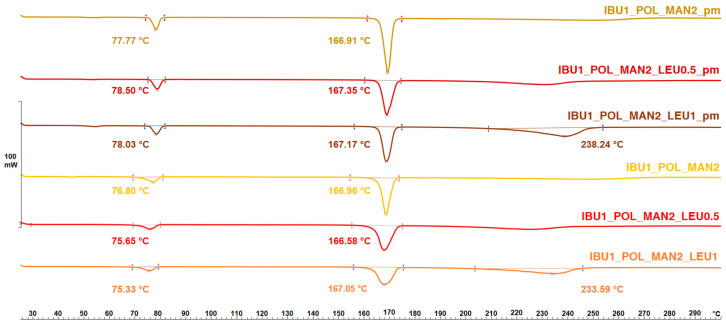
The DSC results of the physical mixtures (IBU1_POL_MAN2_pm, IBU1_POL_MAN2_LEU0.5_pm, IBU1_POL_MAN2_LEU1_pm) and the spray-dried IBU- and MAN-containing samples (IBU1_POL_MAN2_spd, IBU1_POL_MAN2_LEU0.5_spd, IBU1_POL_MAN2_LEU1_spd).

**Figure 6 pharmaceutics-16-01465-f006:**
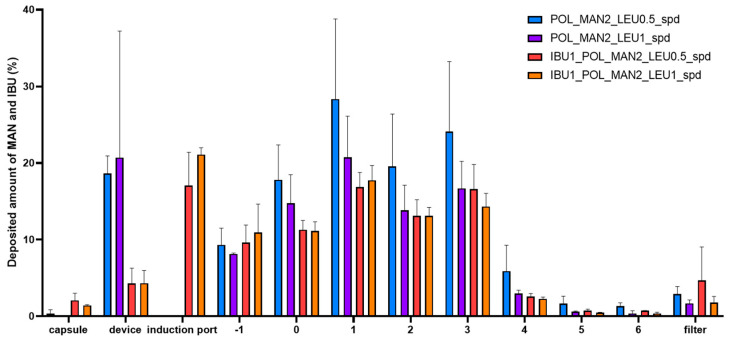
In vitro aerodynamic distribution of the spray-dried samples in the ACI (POL_MAN2_LEU0.5_spd, POL_MAN2_LEU1_spd, IBU1_POL_MAN2_LEU0.5_spd, IBU1_POL_MAN2_LEU1_spd). The data are the means ± SD (*n* = 3 independent measurements).

**Figure 7 pharmaceutics-16-01465-f007:**
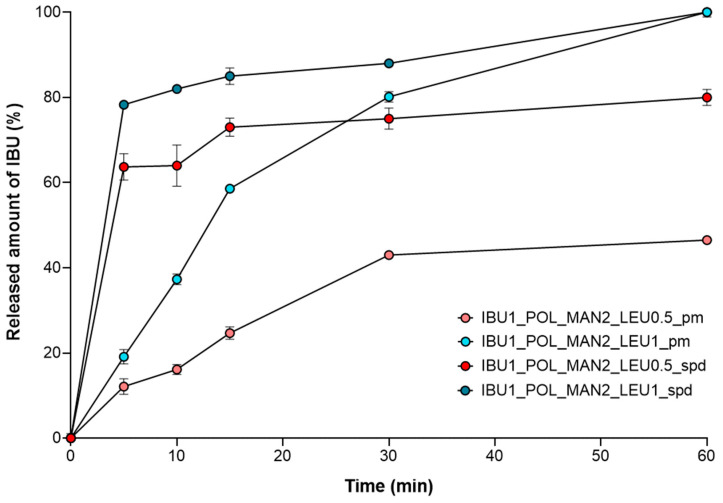
The in vitro dissolution results of the physical mixtures (IBU1_POL_MAN2_LEU0.5_pm, IBU1_POL_MAN2_LEU1_pm) and the prepared IBU- and MAN-containing samples (IBU1_POL_MAN2_LEU0.5_spd, IBU1_POL_MAN2_LEU1_spd). The data are the means ± SD (*n* = 3 independent measurements).

**Figure 8 pharmaceutics-16-01465-f008:**
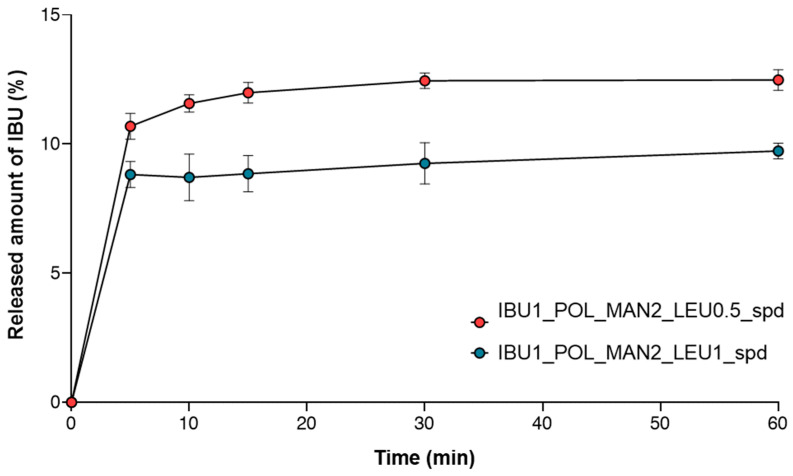
The results of the spray-dried IBU- and MAN-containing samples during the in vitro dissolution test combined with the ACI (IBU1_POL_MAN2_LEU0.5_spd, IBU1_POL_MAN2_LEU1_spd). The data are the means ± SD (*n* = 3 independent measurements).

**Table 1 pharmaceutics-16-01465-t001:** The composition of the spray-dried samples, PMs, and API contents of the IBU-containing samples. The data are the means ± SD (*n* = 3 independent measurements).

Sample	IBU (g)	POL (g)	MAN (g)	LEU (g)	API Content (%)
POL_MAN2_spd	0.00	0.09	2.00	0.00	-
POL_MAN2_LEU0.5_spd	0.00	0.09	2.00	0.50	-
POL_MAN2_LEU1_spd	0.00	0.09	2.00	1.00	-
POL_MAN2_pm	0.00	0.09	2.00	0.00	-
POL_MAN2_LEU0.5_pm	0.00	0.09	2.00	0.50	-
POL_MAN2_LEU1_pm	0.00	0.09	2.00	1.00	-
IBU1_POL_MAN2_spd	1.00	0.09	2.00	0.00	35.17 ± 0.80
IBU1_POL_MAN2_LEU0.5_spd	1.00	0.09	2.00	0.50	27.86 ± 2.43
IBU1_POL_MAN2_LEU1_spd	1.00	0.09	2.00	1.00	24.45 ± 0.31
IBU1_POL_MAN2_pm	1.00	0.09	2.00	0.00	32.36
IBU1_POL_MAN2_LEU0.5_pm	1.00	0.09	2.00	0.50	27.86
IBU1_POL_MAN2_LEU1_pm	1.00	0.09	2.00	1.00	24.45

**Table 2 pharmaceutics-16-01465-t002:** The particle size of the API, the suspension, and the spray-dried samples. The data are the means ± SD (*n*= 3 independent measurements).

Sample	D[0.1] (μm)	D[0.5] (μm)	D[0.9] (μm)	Span	SSA (m^2^/g)
POL_MAN2_spd	3.67 ± 0.08	7.81 ± 0.05	17.77 ± 0.26	1.72 ± 0.05	0.90 ± 0.01
POL_MAN2_LEU0.5_spd	1.54 ± 0.02	2.96 ± 0.05	5.46 ± 0.22	1.32 ± 0.06	2.30 ± 0.01
POL_MAN2_LEU1_spd	1.54 ± 0.003	3.28 ± 0.08	6.66 ± 0.35	1.56 ± 0.07	2.16 ± 0.04
IBU_raw	9.75 ± 8.45	31.78 ± 9.01	95.86 ± 55.12	2.63 ± 1.70	0.43 ± 0.02
IBU_POL_suspension	1.52 ± 0.59	3.39 ± 1.51	17.63 ± 2.85	5.55 ± 2.83	2.32 ± 1.20
IBU1_POL_MAN2_spd	3.15 ± 0.06	8.07 ± 1.99	18.86 ± 1.66	2.04 ± 0.71	0.99 ± 0.05
IBU1_POL_MAN2_LEU0.5_spd	1.43 ± 0.06	3.38 ± 0.13	13.40 ± 5.38	3.51 ± 1.44	2.11 ± 0.13
IBU1_POL_MAN2_LEU1_spd	1.51 ± 0.02	3.28 ± 0.10	6.95 ± 1.14	1.66 ± 0.30	2.18 ± 0.05

**Table 3 pharmaceutics-16-01465-t003:** Density results and powder rheology properties of spray-dried samples. Data are means ± SD (*n* = 3 independent measurements).

Sample	Density (g/cm^3^)	Tapped Density (g/cm^3^)	Hausner Ratio	Carr Index
POL_MAN2_spd	0.12 ± 0.002	0.18 ± 0.008	1.46 ± 0.05	31.67 ± 2.36
POL_MAN2_LEU0.5_spd	0.15 ± 0.002	0.24 ± 0.001	1.65 ± 0.03	39.23 ± 1.09
POL_MAN2_LEU1_spd	0.15 ± 0.052	0.26 ± 0.089	1.74 ± 0.01	42.58 ± 0.39
IBU1_POL_MAN2_spd	0.17 ± 0.011	0.26 ± 0.013	1.49 ± 0.02	32.74 ± 0.84
IBU1_POL_MAN2_LEU0.5_spd	0.23 ± 0.020	0.37 ± 0.024	1.57 ± 0.03	36.22 ± 1.31
IBU1_POL_MAN2_LEU1_spd	0.18 ± 0.008	0.28 ± 0.007	1.59 ± 0.03	37.06 ± 1.34

**Table 4 pharmaceutics-16-01465-t004:** In vitro aerodynamic properties: mass median aerodynamic diameter (MMAD), fine-particle fraction (FPF), and emitted fraction (EF) of spray-dried samples at flow of 60 L/min). Data are means ± SD (*n* = 3 independent measurements).

Sample	MMAD (μm)	FPF (%)	EF (%)
POL_MAN2_LEU0.5_spd	4.42 ± 0.15	70.89 ± 8.59	60.25 ± 4.17
POL_MAN2_LEU1_spd	4.71 ± 0.05	64.34 ± 10.65	56.85 ± 4.03
IBU1_POL_MAN2_LEU0.5_spd	3.88 ± 0.19	63.62 ± 2.99	46.85 ± 0.02
IBU1_POL_MAN2_LEU1_spd	4.76 ± 0.23	61.91 ± 1.55	62.85 ± 0.03

**Table 5 pharmaceutics-16-01465-t005:** In vitro aerodynamic properties determined by using the Spraytec^®^ device of the spray-dried samples at a flow of 60 L/min). The data are the means ± SD (*n* = 3 independent measurements).

Sample	D [0.5] (μm)	Span	SSA (m^2^/g)
POL_MAN2_LEU0.5	4.24 ± 0.36	2.56± 0.66	4.64 ± 0.50
POL_MAN2_LEU1	6.02 ± 0.66	2.81 ± 0.35	3.67 ± 0.47
IBU1_POL_MAN2_LEU0.5	4.16 ± 0.13	3.30 ± 0.72	4.42 ± 0.13
IBU1_POL_MAN2_LEU1	5.12 ± 0.40	3.50 ± 0.52	4.03 ± 0.05

**Table 6 pharmaceutics-16-01465-t006:** In silico deposition in the respiratory system of the samples. The data are the means ± SD (*n* = 3 independent measurements).

Sample	Extrathor. (%)		Bronchial (%)		Acinar (%)		Exhaled (%)	
	5 s	10 s	5 s	10 s	5 s	10 s	5 s	10 s
POL_MAN2_LEU0.5_spd	29.55 ± 7.90	29.17 ± 7.91	16.25 ± 0.06	17.06 ± 0.09	30.93 ± 1.50	34.10 ± 1.91	23.26 ± 6.24	19.67 ± 5.91
POL_MAN2_LEU1_spd	32.55 ± 6.14	32.18 ± 6.22	15.84 ± 3.15	16.59 ± 3.32	28.14 ± 6.31	30.77 ± 6.86	23.48 ± 3.33	20.45 ± 3.96
IBU1_POL_MAN2_LEU0.5_spd	39.99 ± 2.88	39.66 ± 2.89	14.40 ± 0.83	15.07 ± 0.82	26.87 ± 1.49	29.75 ± 1.74	18.74 ± 0.97	15.51 ± 0.78
IBU1_POL_MAN2_LEU1_spd	43.84 ± 4.17	44.83 ± 2.67	14.54 ± 0.89	15.21 ± 0.89	25.88 ± 0.78	27.18 ± 1.03	16.73 ± 4.27	12.79 ± 2.54

## Data Availability

The original contributions presented in the study are included in the article, further inquiries can be directed to the corresponding author.

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
