# Peer review of "Comprehensive In Vitro and In Silico Aerodynamic Analysis of High-Dose Ibuprofen- and Mannitol-Containing Dry Powder Inhalers for the Treatment of Cystic Fibrosis"

_pharmaceutics, 2024, doi:10.3390/pharmaceutics16111465_

Round 1
Reviewer 1 Report
Comments and Suggestions for Authors
The manuscript supplied for review presents a study focused on developing dry powder inhalers (DPIs) containing ibuprofen and mannitol for cystic fibrosis treatment using Poloxamer-188 and leucine as excipients. It highlights successful formulation of spherical particles with suitable aerodynamic properties for lung deposition, offering potential for improved therapeutic delivery with lower systemic exposure and side effects.
I have a few comments that the authors may wish to address
Line 70 - states there is an emerging need for pulmonary NSAID treatments. There should be more discussion of how this method improves upon existing treatments and whether there are other inhaled NSAIDs in development or on the market for similar applications.
Line 111 - the phrase "share a proper water solubility" is not typically used and to my knowledge mannitol and poloxamer are highly soluble in water and leucine is only moderately soluble. Now solubility may not be an issue at the concentrations investigated in this paper but the highlighted statement is not clear.
Lines 57-63 -The manuscript mentions that dry powder inhalers (DPIs) are activated by inhalation, and their effectiveness depends on a sufficient inhalation maneuver. However, cystic fibrosis (CF) patients, especially those with severe lung dysfunction, may not have the required inhalation force to adequately use a DPI. While this is briefly acknowledged, no alternative solutions or mitigation strategies (e.g., for patients with low lung function) are proposed. Plus additional references should be provided for the statements in the lines highlighted above.
The solid-state analyses conducted in the manuscript—XRPD, DSC, SEM, laser diffraction, and density measurements—provide comprehensive insights into the physical and chemical properties of the formulations. These techniques reveal that the spray-dried formulations exhibit partial amorphisation, improved morphological properties for aerosolisation, and reduced crystallinity, all of which contribute to enhancing drug release and lung deposition. However, the formulations prepared in the manuscript have not been subjected to any form of stability analysis which would provide much need information on the effect of relative humidity on the formulations (particularly when mannitol and leucine have been incorporated into some of the formulations as these are hygroscopic materials) . Relative humidity could greatly effect the levels of crystallinity over time as well as issues related to aerosolisation, flowability and particle agglomeration which are necessary to optimise delivery efficiency. Including experimental data or even a discussion would strengthen the manuscript by addressing real-world environmental factors that could influence the effectiveness of the formulations.
Comments on the Quality of English LanguageThroughout the manuscript I have an issue with the use of the term mono-formulation. It is not a traditional term when dealing with a product containing a single API with excipients. It would be more typical to uses terms like "single-component formulation" or "single-drug formulation" to refer to formulations containing only one active pharmaceutical ingredient (API)
"Spray dried" should be hyphenated throughout the document to indicate it is an adjective.
The term "Per os administration" should either be standardised as "oral administration" which is a more common term for consistency with the rest of the text. An alternative would be to italicise "per os" to make it a little clearer
Author Response
Response to Reviewer 1
The manuscript supplied for review presents a study focused on developing dry powder inhalers (DPIs) containing ibuprofen and mannitol for cystic fibrosis treatment using Poloxamer-188 and leucine as excipients. It highlights successful formulation of spherical particles with suitable aerodynamic properties for lung deposition, offering potential for improved therapeutic delivery with lower systemic exposure and side effects.
I have a few comments that the authors may wish to address
Thank you very much for your opinion. Below are listed all the modifications made in the paper, according to the suggestions, and you can find them with purple color in the text.
Line 70 - states there is an emerging need for pulmonary NSAID treatments. There should be more discussion of how this method improves upon existing treatments and whether there are other inhaled NSAIDs in development or on the market for similar applications.
Thank you for your comment. The text was modified.
There is an emerging need for inhaled NSAID medications to reduce inflammation in the lungs beside corticosteroids that have not yet been fulfilled. However, the application of NSAIDs in pulmonary therapy is not common, due to their possible bronchoconstrictive side effect. There are a few NSAID containing DPI in the literature (e.g. IBU and ketoprofen) for CF under development, but no commercially available formulations yet [1–5].
- Malamatari, M.; Somavarapu, S.; Kachrimanis, K.; Buckton, G.; Taylor, K.M.G. Preparation of Respirable Nanoparticle Agglomerates of the Low Melting and Ductile Drug Ibuprofen: Impact of Formulation Parameters. Powder Technol 2017, 308, 123–134, doi:10.1016/j.powtec.2016.12.007.
- Party, P.; Klement, M.L.; Révész, P.S.; Ambrus, R. Preparation and Characterization of Ibuprofen Containing Nano ‐ Embedded ‐ Microparticles for Pulmonary Delivery. Pharmaceutics 2023, 15, 1–16, doi:https://doi.org/10.3390/pharmaceutics15020545.
- Banat, H.; Csóka, I.; Paróczai, D.; Burian, K.; Farkas, Á.; Ambrus, R. A Novel Combined Dry Powder Inhaler Comprising Nanosized Ketoprofen-Embedded Mannitol-Coated Microparticles for Pulmonary Inflammations: Development, In Vitro–In Silico Characterization, and Cell Line Evaluation. Pharmaceuticals 2024, 17, doi:10.3390/ph17010075.
- Stigliani, M.; Aquino, R.P.; Del Gaudio, P.; Mencherini, T.; Sansone, F.; Russo, P. Non-Steroidal Anti-Inflammatory Drug for Pulmonary Administration: Design and Investigation of Ketoprofen Lysinate Fine Dry Powders. Int J Pharm 2013, 448, 198–204, doi:10.1016/j.ijpharm.2013.03.030.
- Yazdi, A.K.; Smyth, H.D.C. Carrier-Free High-Dose Dry Powder Inhaler Formulation of Ibuprofen: Physicochemical Characterization and in Vitro Aerodynamic Performance. Int J Pharm 2016, 511, 403–414, doi:10.1016/j.ijpharm.2016.06.061.
Line 111 - the phrase "share a proper water solubility" is not typically used and to my knowledge mannitol and poloxamer are highly soluble in water and leucine is only moderately soluble. Now solubility may not be an issue at the concentrations investigated in this paper but the highlighted statement is not clear.
Thank you for your comment. The text was modified.
The MAN single-drug formulation was prepared by dissolving MAN, POL and LEU in different concentrations. These components share a proper water solubility and mix well in the applied concentrations (Table 1.).
Lines 57-63 -The manuscript mentions that dry powder inhalers (DPIs) are activated by inhalation, and their effectiveness depends on a sufficient inhalation maneuver. However, cystic fibrosis (CF) patients, especially those with severe lung dysfunction, may not have the required inhalation force to adequately use a DPI. While this is briefly acknowledged, no alternative solutions or mitigation strategies (e.g., for patients with low lung function) are proposed. Plus additional references should be provided for the statements in the lines highlighted above.
Thank you for your comment. The text was modified.
To achieve optimal lung deposition with DPIs both a sufficient inhalation maneuver and the optimal type of DPI device are required. The inhalators, classified as a low-resistance inhalator (e.g. Breezhaler® single-dose device), need a weaker inspiratory effort to achieve high flow rates across the device even in case of patients with low lung function [6,7].
- Molimard, M.; Kottakis, I.; Jauernig, J.; Lederhilger, S.; Nikolaev, I. Performance Characteristics of Breezhaler® and Aerolizer® in the Real-World Setting. Clin Drug Investig 2021, 41, 415–424, doi:10.1007/s40261-021-01021-w.
- Clark, A.R.; Weers, J.G.; Dhand, R. The Confusing World of Dry Powder Inhalers: It Is All about Inspiratory Pressures, Not Inspiratory Flow Rates. J Aerosol Med Pulm Drug Deliv 2020, 33, 1–11, doi:10.1089/jamp.2019.1556.
The solid-state analyses conducted in the manuscript—XRPD, DSC, SEM, laser diffraction, and density measurements—provide comprehensive insights into the physical and chemical properties of the formulations. These techniques reveal that the spray-dried formulations exhibit partial amorphisation, improved morphological properties for aerosolisation, and reduced crystallinity, all of which contribute to enhancing drug release and lung deposition. However, the formulations prepared in the manuscript have not been subjected to any form of stability analysis which would provide much need information on the effect of relative humidity on the formulations (particularly when mannitol and leucine have been incorporated into some of the formulations as these are hygroscopic materials) . Relative humidity could greatly effect the levels of crystallinity over time as well as issues related to aerosolisation, flowability and particle agglomeration which are necessary to optimise delivery efficiency. Including experimental data or even a discussion would strengthen the manuscript by addressing real-world environmental factors that could influence the effectiveness of the formulations.
Thank you for your comment. The text was modified.
Furthermore, MAN is an attractive excipient, due to being less hygroscopic than some of the other sugars and not absorbing moisture until the RH is over 90%, which is beneficial in terms of better physical and chemical stability of the DPI formulation [8–11].
A further excipient, L-Leucine (LEU), was applied to enhance the dispersity of the particles, thereby improving the aerosolization and decreasing the hygroscopicity of the powder[12,13]. Moreover, LEU also has stability-enhancing effect, mitigating moisture absorption with the hydrophobic shell [14,15].
- Mönckedieck, M.; Kamplade, J.; Fakner, P.; Urbanetz, N.A.; Walzel, P.; Steckel, H.; Scherließ, R. Dry Powder Inhaler Performance of Spray Dried Mannitol with Tailored Surface Morphologies as Carrier and Salbutamol Sulphate. Int J Pharm 2017, 524, 351–363, doi:10.1016/j.ijpharm.2017.03.055.
- Kaialy, W.; Hussain, T.; Alhalaweh, A.; Nokhodchi, A. Towards a More Desirable Dry Powder Inhaler Formulation: Large Spray-Dried Mannitol Microspheres Outperform Small Microspheres. Pharm Res 2014, 31, 60–76, doi:10.1007/s11095-013-1132-2.
- Li, X.; Vogt, F.G.; Hayes, D.; Mansour, H.M. Design, Characterization, and Aerosol Dispersion Performance Modeling of Advanced Spray-Dried Microparticulate/Nanoparticulate Mannitol Powders for Targeted Pulmonary Delivery as Dry Powder Inhalers. J Aerosol Med Pulm Drug Deliv 2014, 27, 81–93, doi:10.1089/jamp.2013.1078.
- Kaialy, W.; Martin, G.P.; Ticehurst, M.D.; Momin, M.N.; Nokhodchi, A. The Enhanced Aerosol Performance of Salbutamol from Dry Powders Containing Engineered Mannitol as Excipient. Int J Pharm 2010, 392, 178–188, doi:10.1016/j.ijpharm.2010.03.057.
- Li, L.; Sun, S.; Parumasivam, T.; Denman, J.A.; Gengenbach, T.; Tang, P.; Mao, S.; Chan, H.K. L-Leucine as an Excipient against Moisture on in Vitro Aerosolization Performances of Highly Hygroscopic Spray-Dried Powders. European Journal of Pharmaceutics and Biopharmaceutics 2016, 102, 132–141, doi:10.1016/j.ejpb.2016.02.010.
- Feng, A.L.; Boraey, M.A.; Gwin, M.A.; Finlay, P.R.; Kuehl, P.J.; Vehring, R. Mechanistic Models Facilitate Efficient Development of Leucine Containing Microparticles for Pulmonary Drug Delivery. Int J Pharm 2011, 409, 156–163, doi:10.1016/j.ijpharm.2011.02.049.
- Alhajj, N.; O’Reilly, N.J.; Cathcart, H. Leucine as an Excipient in Spray Dried Powder for Inhalation. Drug Discov Today 2021, 26, 2384–2396, doi:10.1016/j.drudis.2021.04.009.
- Dieplinger, J.; Isabel Afonso Urich, A.; Mohsenzada, N.; Pinto, J.T.; Dekner, M.; Paudel, A. Influence of L-Leucine Content on the Aerosolization Stability of Spray-Dried Protein Dry Powder Inhalation (DPI). Int J Pharm 2024, 666, doi:10.1016/j.ijpharm.2024.124822.
Comments on the Quality of English Language
Throughout the manuscript I have an issue with the use of the term mono-formulation. It is not a traditional term when dealing with a product containing a single API with excipients. It would be more typical to uses terms like "single-component formulation" or "single-drug formulation" to refer to formulations containing only one active pharmaceutical ingredient (API)
Thank you for your comment. The text was modified. Single-drug product or single-drug formulation terms were used in the text instead of mono-fomulation.
"Spray dried" should be hyphenated throughout the document to indicate it is an adjective.
Thank you for your comment. The text was modified. Spray-dried term was used in the text.
The term "Per os administration" should either be standardised as "oral administration" which is a more common term for consistency with the rest of the text. An alternative would be to italicise "per os" to make it a little clearer
Thank you for your comment. The text was modified. Per os term is now italicized.

Reviewer 2 Report
Comments and Suggestions for Authors
The manuscript presents preparation and characterisation of dry powder inhaler formulations of Ibuprofen with mannitol, Poloxamer 188 and Leucine. Overall, the approach presented in this work is not new because the formulation technique is already well developed. There is no clear justification for the advantage of Ibuprofen DPI in CF management. The formulation and characterisation presented by the authors are basic and the manuscript lacks clarity with several grammatical and spelling errors. There is no adequate discussion of the obtained results. Therefore the manuscript needs a thorough major revision before it can be accepted for publication.
1. In the introduction, discuss the current approach to treating CF. E.g. Trikafta is a wonder drug that is making a huge difference in CF treatment and management worldwide. However, it is suitable in those who have at least one copy of the F508del mutation in the CFTR gene or another mutation that is responsive to treatment with the drug.
2. Explain more about Poloxamer 188 in the introduction, highlighting why it was selected, if it is approved for use as an excipient in inhaled formulations and if it is safe to be administered to the lungs.
3. Line 110: Check the spelling ‘solving’
4. Line 133: Mention pump rate in terms of mL/min
5. Line 206-207: ‘Span 85 and 206 cyclohexane (1+99 w/w%) mixture’ – Does 1+99 mean 1 part of span 85 and 99 parts of 206 cyclohexane?
6. Lines 286-290: Requires correction for use of English, grammar and spelling
7. Line 291: ‘In the case of POL_MAN2_LEU0.5 and 291 POL_MAN2_LEU1.’ Incomplete sentence? The sentence following this one is also incomplete.
8. Line 329: XRD is not structure analysis in this case
9. Section 4: The subheading is discussion but it looks like a conclusion section.
Comments on the Quality of English LanguageQuality of English Language in this manuscript is poor. Thorough revision is recommended.
Author Response
Response to Reviewer 2
The manuscript presents preparation and characterisation of dry powder inhaler formulations of Ibuprofen with mannitol, Poloxamer 188 and Leucine. Overall, the approach presented in this work is not new because the formulation technique is already well developed. There is no clear justification for the advantage of Ibuprofen DPI in CF management. The formulation and characterisation presented by the authors are basic and the manuscript lacks clarity with several grammatical and spelling errors. There is no adequate discussion of the obtained results. Therefore the manuscript needs a thorough major revision before it can be accepted for publication.
Thank you very much for your opinion. Below are listed all the modifications made in the paper, according to the suggestions, and you can find them with orange color in the text.
- In the introduction, discuss the current approach to treating CF. E.g. Trikafta is a wonder drug that is making a huge difference in CF treatment and management worldwide. However, it is suitable in those who have at least one copy of the F508del mutation in the CFTR gene or another mutation that is responsive to treatment with the drug.
Thank you for your comment. The text was modified.
The primary aim of CF therapy is to maintain or even enhance the patients' quality of living. Alternative forms of treatment include specific diets and airway clearance treatments, which do not involve the use of pharmaceuticals. Mucolytics, such as mannitol, dornase-alpha, and hypertonic sodium chloride solution, are also commonly utilized to dilute the thick mucus. Bacterial infections potentially leading to exacerbations can be treated with antibiotics. The most typical ways to provide these medications (such as azithromycin, colistin, tobramycin, and ciprofloxacin) are by intravenous, pulmonary, and oral routes. In addition to antibiotics anti-inflammatory drugs, particularly corticosteroids, are helpful in the treatment; nevertheless, their side effects could be harmful in long-term therapies. CFTR modulators, such as Symdeko Trikafta, Orkambi, and Kalydeco, have significantly improved CF management and therapy. However, they are only suitable in specific mutations in the CFTR gene. These are the priciest medications, and before administering them to patients, a genetic examination is essential [16–18].
- DeSimone, E.; Tilleman, J.; Giles, M.E.; Moussa, B. Cystic Fibrosis: Update on Treatment Guidelines and New Recommendations. U.S. Pharmacist 2018, 43, 16–21.
- Mogayzel, P.J.; Naureckas, E.T.; Robinson, K.A.; Mueller, G.; Hadjiliadis, D.; Hoag, J.B.; Lubsch, L.; Hazle, L.; Sabadosa, K.; Marshall, B. Cystic Fibrosis Pulmonary Guidelines: Chronic Medications for Maintenance of Lung Health. Am J Respir Crit Care Med 2013, 187, 680–689, doi:10.1164/rccm.201207-1160OE.
- Rafeeq, M.M.; Murad, H.A.S. Cystic Fibrosis: Current Therapeutic Targets and Future Approaches. J Transl Med 2017, 15, 1–9, doi:10.1186/s12967-017-1193-9.
- Explain more about Poloxamer 188 in the introduction, highlighting why it was selected, if it is approved for use as an excipient in inhaled formulations and if it is safe to be administered to the lungs.
Thank you for your comment. The text was modified.
As additive, we used Poloxamer-188 (POL) polymer, which is safe at a low dose and can help to create porous particles [19,20]. Moreover, POL coatings present on the particle surface could diminish the extent of biophysical inhibition of lung surfactant [21].
- Pilcer, G.; Amighi, K. Formulation Strategy and Use of Excipients in Pulmonary Drug Delivery. Int J Pharm 2010, 392, 1–19, doi:10.1016/j.ijpharm.2010.03.017.
- Lindenberg, F.; Sichel, F.; Lechevrel, M.; Respaud, R.; Guillaume, S.-L. Evaluation of Lung Cell Toxicity of Surfactants for Inhalation Route. Journal of Toxicology and Risk Assessment 2019, 5, doi:10.23937/2572-4061.1510022ï.
- Beck-Broichsitter, M.; Bohr, A.; Ruge, C.A. Poloxamer-Decorated Polymer Nanoparticles for Lung Surfactant Compatibility. Mol Pharm 2017, 14, 3464–3472, doi:10.1021/acs.molpharmaceut.7b00477.
- Line 110: Check the spelling ‘solving’
Thank you for your comment. The text was modified.
The MAN single-drug formulation was prepared by dissolving MAN, POL and LEU in different concentrations.
- Line 133: Mention pump rate in terms of mL/min
Thank you for your comment. The text was modified.
Based on our preliminary experiments, the spray drying settings were the following: inlet temperature: 70 °C, aspirator capacity: 85%, airflow rate: 500 L/h, pump rate: 3 ml/min.
- Line 206-207: ‘Span 85 and 206 cyclohexane (1+99 w/w%) mixture’ – Does 1+99 mean 1 part of span 85 and 99 parts of 206 cyclohexane?
Thank you for your comment. The text was modified. The number 206 was not part of the text, it was the number of the line.
To simulate the pulmonary adhesive circumstances, the collection plates on the stages were coated with Span 85 and cyclohexane (1+99 w/w%) mixture.
- Lines 286-290: Requires correction for use of English, grammar and spelling
Thank you for your comment. The text was modified.
After spray drying, the D[0.5] values of the samples were between 3–8 μm. The result met the requirements of the pulmonary powders in the case of LEU containing products. The particle size is one of the most important factors in achieving lung deposition. Incorporating LEU, the geometric size of the spray-dried particles decreased, which led to an increasing SSA.
- Line 291: ‘In the case of POL_MAN2_LEU0.5 and 291 POL_MAN2_LEU1.’ Incomplete sentence? The sentence following this one is also incomplete.
Thank you for your comment. The text was modified. The incomplete sentence was deleted.
- Line 329: XRD is not structure analysis in this case
Thank you for your comment. The title was modified to: Results of the crystal structure analysis using X-ray powder diffraction.
- Section 4: The subheading is discussion but it looks like a conclusion section.
Thank you for your comment. The text was modified.
This study assessed that a single-drug-DPI preparation containing MAN and an innovative DPI combining MAN and IBU was developed using appropriate excipients for the treatment of cystic fibrosis. POL promoted the creation of more porous particles, and it could be beneficial to the reduction of the negative effects of the surfactant. The application of MAN, besides its mucus diluting effect, led to a preferable spherical form. Using LEU, the particle surface became rough and more hydrophobic, which helped the aerodynamic properties and can lead to improved stability. The results seemed con-sistent with the initial aim that a high drug loading can be achieved, which can result in high dose drug delivery to the lung. A comprehensive physicochemical and dosage form investigation was executed. The particle size of the products was in the range we pre-dicted, the diameter of the formulations containing LEU was under 5 μm to target the lungs (POL_MAN2_LEU0.5_spd: 2.96 ± 0.05 μm; POL_MAN2_LEU1_spd: 3.28 ± 0.08 μm; IBU_POL_MAN2_LEU0.5_spd: 3.38 ± 0.13 μm; POL_MAN2_LEU1_spd: 3.28 ± 0.10 μm) according to the laser diffraction analysis. The only MAN containing formulation showed a spherical form, the IBU containing formulations appeared porous on the SEM pictures. Our findings suggest that this unique, porous shape was caused by the com-bined use of IBU and POL. Furthermore, the aerodynamic characteristics were found to be sufficient. MMAD results were also in the pulmonary required range (POL_MAN2_LEU0.5_spd: 4.42 ± 0.15 μm; POL_MAN2_LEU1_spd: 4.71 ± 0.05 μm; IBU1_POL_MAN2_LEU0.5_spd: 3.88 ± 0.19 μm; IBU1_POL_MAN2_LEU1_spd: 4.76 ± 0.23 μm). FPF values were large in comparison to the commercially available products (POL_MAN2_LEU0.5_spd: 70.89 ± 8.59 %; POL_MAN2_LEU1_spd: 64.34 ± 10.65 %; IBU1_POL_MAN2_LEU0.5_spd: 63.62 ± 2.99 %; IBU1_POL_MAN2_LEU1_spd: 61.91 ± 1.55 %). The measurements with the laser diffraction based Spraytec® device clarified, that the lower LEU containing product showed preferable aerodynamic particle size for targeting the lung (POL_MAN2_LEU0.5: 4.24 ± 0.36 μm; POL_MAN2_LEU1: 6.02 ± 0.66 μm; IBU1_POL_MAN2_LEU0.5: 4.16 ± 0.13 μm; IBU1_POL_MAN2_LEU1: 5.12 ± 0.40 μm). In agreement with our original concept, the micronized IBU-containing samples dissolved more quickly than the raw material. According to the stochastic lung model, a large percentage of the formulations reached the bronchial and acinar parts of the lung (between 42 and 51%) and the prolonged breath holding time decreased the amount of the exhaled fraction. To examine the impact of varying relative humidities and tem-peratures on the crystalline structure of the APIs and excipients, it will be crucial to carry out a stability test of the presented DPIs in the future. In addition to having a therapeutic effect, the composition of our single-drug-preparation can serve as a base for DPI formulations developed with other active ingredients in the future. Our results can contribute to expanding our knowledge on the high dose DPI formulations, thereby providing the basis for future research of DPI formulations for the potential treatment of cystic fibrosis patients.
Comments on the Quality of English Language
Quality of English Language in this manuscript is poor. Thorough revision is recommended.
Thank you for your comment. The manuscript was revised.

Round 2
Reviewer 2 Report
Comments and Suggestions for Authors
Comments on revised version of manuscript pharmaceutics-3262041:
1. The correction regarding the X-ray analysis are not convincing. Under both method (section 2.2.8) and result (section 3.4) the authors discuss about crystal structural analysis/investigation. There is no data provided on the crystal structures. Authors have provided and discussed X-ray diffractograms which only inform on either presence or absence of crystalline particles in the sample.
2. The discussion section is poor and lacks robust discussion of the results obtained. There is no specific concluding remarks.
Comments on the Quality of English LanguageQuality of English Language in this manuscript is average.
Author Response
I appreciate your thoughtful revision. The changes are listed below.
- The correction regarding the X-ray analysis are not convincing. Under both method (section 2.2.8) and result (section 3.4) the authors discuss about crystal structural analysis/investigation. There is no data provided on the crystal structures. Authors have provided and discussed X-ray diffractograms which only inform on either presence or absence of crystalline particles in the sample.
Thank you for your comment. Section 2.2.8. and 3.4. were modified.
For the characterization of the crystalline materials, X-ray powder diffraction (XRPD) spectra were recorded with the help of the BRUKER D8 Advance X-ray dif-fractometer (Bruker AXS GmbH, Karlsruhe, Germany).
Crystallinity results of the particles determined by X-ray powder diffraction
Overall, the spray drying procedures decreased the number of crystallin particles of MAN and IBU.
- The discussion section is poor and lacks robust discussion of the results obtained. There is no specific concluding remarks.
Thank you for your comment. The text was modified.
This study assessed that a single-drug-DPI preparation containing MAN and an innovative DPI combining MAN and IBU was developed using appropriate excipients for the treatment of CF. POL promoted the creation of more porous particles, and it could be beneficial to the reduction of the negative effects of the surfactant. The application of MAN, besides its mucus diluting effect, led to a preferable spherical form. Using LEU, the particle surface became rough and more hydrophobic, which helped the aerodynamic properties and led to improved stability. In the case of IBU, application of high dose (2–3 mg/kg/dose twice daily) is recommended for local CF treatment. According to the outcome of the wet milling and the spray drying, the formulation seemed consistent with the initial aim that a high drug loading can be achieved, which can result in high dose drug delivery to the lung.
A comprehensive physicochemical and dosage form investigation was executed to determine the pulmonary applicability of the prepared samples. The particle size of the products was in the range we predicted, the diameter of the formulations containing LEU was under 5 μm to target the lungs (POL_MAN2_LEU0.5_spd: 2.96 ± 0.05 μm; POL_MAN2_LEU1_spd: 3.28 ± 0.08 μm; IBU_POL_MAN2_LEU0.5_spd: 3.38 ± 0.13 μm; POL_MAN2_LEU1_spd: 3.28 ± 0.10 μm) according to the laser diffraction analysis. The only MAN containing formulation showed a spherical form, and the IBU containing formulations appeared porous on the SEM pictures. Our findings suggest that this unique, porous shape was caused by the combined use of IBU and POL. Furthermore, the aerodynamic characteristics were found to be sufficient. MMAD results of the LUE containing formulations were also in the required range for pulmonary delivery (POL_MAN2_LEU0.5_spd: 4.42 ± 0.15 μm; POL_MAN2_LEU1_spd: 4.71 ± 0.05 μm; IBU1_POL_MAN2_LEU0.5_spd: 3.88 ± 0.19 μm; IBU1_POL_MAN2_LEU1_spd: 4.76 ± 0.23 μm). FPF values were larger in comparison to the commercially available products in a Breezehaler® device (POL_MAN2_LEU0.5_spd: 70.89 ± 8.59 %; POL_MAN2_LEU1_spd: 64.34 ± 10.65 %; IBU1_POL_MAN2_LEU0.5_spd: 63.62 ± 2.99 %; IBU1_POL_MAN2_LEU1_spd: 61.91 ± 1.55 %). The measurements with the laser diffraction-based Spraytec® device equipped with an inhalation cell, clarified that the lower LEU containing product showed a preferable aerodynamic particle size for targeting the lung (POL_MAN2_LEU0.5: 4.24 ± 0.36 μm; POL_MAN2_LEU1: 6.02 ± 0.66 μm; IBU1_POL_MAN2_LEU0.5: 4.16 ± 0.13 μm; IBU1_POL_MAN2_LEU1: 5.12 ± 0.40 μm). In agreement with our original concept, from the micronized IBU-containing samples the drug dissolved more quickly (69±3.09% from the IBU1_POL_MAN2_LEU0.5, 78.29±0.69% from the IBU1_POL_MAN2_LEU1 within the first 5 minutes) than from the raw material containing physical mixtures due to the decreased particle size of the drug reached by high-performance wet milling. In accordance with the ACI and Spraytec® results, the combined dissolution test also proved that the IBU1_POL_MAN2_LEU0.5 is the more beneficial formulation in comparison to IBU1_POL_MAN2_LEU1. Using the in silico method, we predicted the deposition of the formulations in human airways. According to the stochastic lung model, a large percentage of the formulations reached the bronchial and acinar parts of the lung (between 42 and 51%) and the prolonged breath holding time decreased the amount of the exhaled fraction. To examine the impact of varying relative humidities and temperatures on the crystalline structure of the APIs and excipients, it will be crucial to carry out a stability test of the presented DPIs in the future. In addition to having a therapeutic effect, the composition of our single-drug preparation can serve as a base for DPI formulations developed with other APIs in the future.
In conclusion, the combination of various excipients and APIs was observed, particularly from the perspective of particle size and shape. A comprehensive aerodynamic analysis was presented, which included various in vitro studies to measure the aerodynamic diameter and evaluate drug dissolution as well as an in silico method to validate our predictions for the human body. The results can contribute to expanding our knowledge on the high dose DPI formulations, thereby providing the basis for future research of DPI formulations for the potential treatment of CF patients.
